# Thermal Discomfort Levels, Building Design Concepts, and Some Heat Mitigation Strategies in Low-Income Communities of a South Asian City

**DOI:** 10.3390/ijerph18052535

**Published:** 2021-03-04

**Authors:** Sana Ehsan, Farhat Abbas, Muhammad Ibrahim, Bashir Ahmad, Aitazaz A. Farooque

**Affiliations:** 1Department of Environmental Sciences and Engineering, Government College University Faisalabad, Faisalabad 38000, Pakistan; sana.env@live.com; 2School of Climate Change and Adaptation, University of Prince Edward Island, Charlottetown, PE C1A 4P3, Canada; 3Climate Energy and Water Research Institute, Pakistan Agricultural Research Council, Islamabad 44000, Pakistan; dr.bashir70@gmail.com; 4Faculty of Sustainable Design Engineering, University of Prince Edward Island, Charlottetown, PE C1A 4P3, Canada

**Keywords:** climate change, heat stress, congested areas, ventilation, arid regions

## Abstract

Heat stress provokes thermal discomfort to people living in semiarid and arid climates. This study evaluates thermal discomfort levels, building design concepts, and some heat mitigation strategies in low-income neighborhoods of Faisalabad, Pakistan. The outdoor and indoor weather data are collected from April to August 2016 using a weather station installed ad hoc in urban settings, and the 52 houses of the five low-income participating communities living in congested and less environment-friendly areas of Faisalabad. The *discomfort index* values, related to the building design concepts, including (i) house orientation to sunlight and (ii) house ventilation, are calculated from outdoor and indoor dry-bulb and wet-bulb temperatures. Our results show that although June was the hottest month of summer 2016, based on the monthly mean temperature of the Faisalabad region, the month of May produced the highest discomfort levels, which were higher in houses exposed to sunlight and without ventilation. The study also identifies some popular heat mitigation strategies adopted by the five participating low-income communities during various heat-related health complaints. The strategies are gender-biased and have medical, cultural/customary backgrounds. For example, about 52% of the males and 28% of the females drank more water during dehydration, diarrhea, and eye infection. Over 11% and 19% of the males and females, respectively, moved to cooler places during fever. About 43% of the males and 51% of the females took water showers and rested to combat flu (runny nose), headache, and nosebleed. The people did not know how to cure muscular fatigue, skin allergy (from a type of Milia), and mild temperature. Planting trees in an area and developing open parks with greenery and thick canopy trees can be beneficial for neighborhoods resembling those evaluated in this study.

## 1. Introduction

There has been a rise of 0.74 °C in the global average temperature during the last 100 years—the latter half of which experienced extreme temperatures and intense precipitations with an estimated rise to 1.8–4.0 °C (estimated average of 3.0 °C) by 2100 [1]. The weather of the arid and semiarid zone countries is specifically inclined to heat stress. The low-income people of these countries face harsh weather challenges with a probability of deadly events [2]. Heat stress is considered as the key health issue of people living in low-income areas of Asian cities as declared by IPCC (the Intergovernmental Panel on Climate Change) [3]. For example, the heat stress during 2015 resulted in mortalities in India and Pakistan. Relationships have been established between temperature and mortality, with a strong link between weather and the fitness of people [4]. Mitigation strategies are conventionally derived from the common outdoor meteorological information, such as temperature and humidity, generally used to produce warning signs for heatwaves [5].

Pakistan is among South Asian countries with its numerous regions having arid and semiarid climates. The country is categorized as the world’s most climate-vulnerable region [6]. Heat stress warnings are commonly issued during the summer days in Pakistan. The local outdoor temperature causes people’s exposure to heat stress [7]. Poor living conditions (i.e., no air conditioners to run during summers) and improper design of the living places, such as buildings without vents and built with no plans, by an architect, strongly affect the indoor environment [8,9]. The buildings of small size or without windows act like caves [10]. The urban heat island effect extends the sufferings of the people from heat stress, particularly in congested low-income neighborhoods that cannot afford air conditioning to combat harsh weather [11,12]. The low-income communities in major cities of Pakistan, including Faisalabad live under congested conditions. Their houses do not have proper building design concepts. They are built with poor ventilation and unavoidable orientation to the sun. Most of the houses in Pakistan face South and/or East to avoid sun orientation during summer. North–East and South–East oriented houses/windows/vents are recommended for ventilation. However, the low-income communities do not have control over choosing the orientation of their houses because of limited available places. House orientation to the sun exacerbates indoor temperature and humidity. These conditions lead to cause a high feel-like effect (of temperature) for dwellers during the early summer of humid days that worsen during peak-temperature hours of the day. The feel-like effect is not the real temperature and is based on air temperature, wind speed, and humidity [13]. The phenomenon of the feel-like effect is temperature and/or thermal sensation felt by a human body being hot or cold. Once exposed to winds, human skin becomes colder than the real temperature because the winds draw away body heat with sweat. Therefore, no perspiration during humid conditions keeps the human body hotter than the real temperature.

In arid or semiarid zones, the climates with little to no humidity in the air, allow solar radiation to penetrate the earth without any interference [14]. In such situations, the earth’s surface heats up during the day because of solar radiations and cools down during the night because nothing blocks or traps the heat generated during the day [13]. Coastal entities, such as Karachi in Pakistan, face heatwaves that result from interactions of land-ocean-atmosphere processes causing increases in humidity and temperature [15]. Climate change is expected to exacerbate the frequency, intensity, and duration of heatwaves [16].

People living indoors or working outdoor face exposure to heat hazards in low-income neighborhoods of South Asian cities. Exhaust from the air conditioners, as well as buildings without air conditioning, both lead to the conditions of severe heat stress outdoor and indoors. Night-time indoor temperatures stand important because of the value of this time when residents of a building respite from the harsh hot day. The interactions of indoor temperature and humidity result in thresholds for heat stress facing the dwellers [17], resulting in heat-related morbidity and mortality, heat stress, ground-level ozone, and undernutrition [18]. The thermal thresholds of a certain value have been reported to cause health risks. In Barcelona, Spain, on higher humid days, it was found that a 2 °C higher temperature than the threshold (23 °C) accompanied with relative humidity above 85% potentially caused health risks. Thresholds have been reported riskier for elderly people than youngsters [19], and measured in terms of *discomfort index*—an index for measuring the discomfort felt in warm weather because of the combined effects of ambient air temperature and humidity of the air. High discomfort level, caused by heat waves, adversely affects human well-being [20]. Especially, high night-time temperatures disturb sleep, adversely affect next-day activities, and even to the extent of mortality in arid or semiarid climate zone [21,22].

Nicol and Roaf [23] explained the concepts of understanding behind the thermal relationship between people and buildings, socio-economic aspects [24], psychology [25], physiology [26] which underlies the related health effects and the ways to interpret the body’s response to the environment either as comfort or discomfort [27]. Worldwide, the varying cultural and architectural setups have increasing pressures to exacerbate the challenge of comfortable living [23]. A research question for this study was that how the outdoor weather conditions and building design concepts raise the indoor discomfort levels. Several reports exist on thermal comfort faced by the people living inside modern buildings (see Nicol and Roaf [23], and the references therein). However, thermal discomfort levels, building design concepts, and some heat mitigation strategies in low-income neighborhoods of cities in arid/semiarid zones of South Asia remained unreported. Therefore, this study evaluated thermal discomfort levels, building design concepts, and some heat mitigation strategies in low-income neighborhoods of Faisalabad, Pakistan.

## 2. Materials and Methods

Planning for this study started during a seminar held at the Government College University Faisalabad, Pakistan. The theme of this gathering was to create awareness about heat stress mitigation techniques. These activities were in the wake of the June 2015 Karachi heatwaves, causing deaths of over 700 people from dehydration and heatstroke [28]. Faisalabad was chosen as a study site because of having Karachi resembling slums and congested neighborhoods of low-income communities. Another edge of this study location was the availability of the basic instrumentations from projects of the Higher Education Commission of Pakistan and the Himalayan Adaptation, Water and Resilience (HI-AWARE) consortium supported by the Collaborative Adaptation Research Initiative in Africa and Asia. A student (the lead author) was hired to conduct this research for her Ph.D. dissertation supervised by the second author of this report.

The project team designed a questionnaire and held awareness seminars in the low-income communities of Faisalabad (Figure 1). After the seminars, the heads of some willing households were approached through the cooperative community leaders who were educated about the study’s importance and convinced for cooperation. A general survey questionnaire was shared with the potential participant about data to be collected for this study. Some of the questions, leading to privacy disclosure, were simplified on the demand of the participants. The final questions included socio-economic conditions, family information, health-related issues, and customary heat mitigation strategies adopted by the dwellers to combat heat-related complaints. Access to the personal/family rooms of the participants was granted on the terms and conditions to let female students only collect indoor data and avoid taking photos of the indoor environments.

### 2.1. Study Sites

This study was conducted in Faisalabad, which is the third-largest city in Pakistan. The population of Faisalabad is about eight million. Faisalabad features a semiarid climate (BWh) in Köppen-Geiger classification with very hot and humid summers and cool, dry winters. This city is situated on the plains of the northeast of Punjab with an elevation of 184 m above mean sea level at longitude 73–74° East, and latitude 30–31.5° North. The study site comprised five low-income neighborhoods in the western and central part of the city; (i) the clock tower district, which is the city center comprising old architecture buildings, (ii) D block of Gulam-Muhammad Abad, which is characterized by more vegetation like trees and shrubs and wide roads, (iii) Gulam-Muhammad Abad, a congested area with few trees but shades from buildings, (iv) Muradabad–a slum alongside the city’s main drain, and (v) Babuwala—a slum across main Jhang Road, densely populated and with little vegetation (Figure 2). A historical record of monthly means record of the past 50-years weather data for rainfall, minimum temperature, and maximum temperature for the Faisalabad region is shown in Figure 2.

The participants of this study were low-income communities of Faisalabad who work as a laborer on a daily basis and whose literacy rate is below 10%. As depicted in the participating communities’ photos, taken with permission, their living standards are below the international poverty lines set by the United Nations (Figure 3). They were reluctant to mention it, but their daily income was less than $2 US. One out of the 6–7 members of a family worked for a living, and the rest were his/her dependents. Convincing such low-literate and low-income communities to participate in this study was challenging as there were no financial incentives to offer them for their cooperation. Additionally, in a Muslim and cultural setup of South Asia, permissions to enter their private/personal living places were hard to get manage for the continuous period of five months without any promises to help them ease their living conditions, even to the extent of buying them a pedestal fan with a minimal cost of $100 US. Figure 3 presents snapshots of the neighborhood (left) and indoor conditions (right) of (**a**) Site 1: The clock tower district; (**b**) Site 2: D block of Gulam-Muhammad Abad; (**c**) Site 3: Gulam-Muhammad Abad; (**d**) Site 4: Muradabad, and (**e**) Site 5: Babuwala.

### 2.2. Households and Housing Building Design Concepts Data

With the help of community heads, 52 sample houses were selected within the five low-income participating communities. Detailed household data were also collected by interviewing the house owners individually. The survey questionnaire designed to collect human behavior data comprised questions about health conditions specific to heat, as well as questions based on socio-economic parameters. The regular questionnaire was done on a monthly basis. The socio-economic parameters included the weekly income of the entire family, monthly electricity bill, and power cut-off hours they are suffering. This questionnaire included the sections like how they are coping with heat stress and what type of health issues they are facing gender-wise.

The questionnaire included questions like (i) households’ ratio (i.e., number of persons per house), (ii) gender proportion (female to male ratio), (iii) crowding ratio (the number of rooms per persons), (iv) building design concept (i.e., if houses with proper ventilation, such as windows and/or vents, and if the houses were exposed to sunlight versus a house with shades of trees, the neighboring building, or an upper story construction), (v) possession of necessities, such as a fan, washing machine, evaporative cooler, air conditioner, or fridge, and (vi) experience of a daily power outage. Description of variables and summary statistics of socio-economic conditions and building design concepts of the house considered in this study are given in Table 1.

The average family size of the surveyed communities was 6.6 persons per household. There were congested situations making houses in the study area overcrowded as judged from the ratio between the number of rooms and people per house, e.g., 292 people/114 rooms = 2.56. Most of the houses had common building materials, such as tiles in the roof and bricks in walls. Rooftops were made from tiles, and walls were made up of bricks. Although all households owned one fan (ceiling and/or pedestal), power shortage was a permanent problem in the whole study area.

Houses of all the five communities had no proper design concepts. For example, the rooms did not have poor ventilation (on the two opposite walls). Over half of the rooms (~32) had one or more windows, and the rest (~20) were without a window. Half of the rooms had vents, and <10 of the rooms had two or more, but improperly placed vents. About 90% of the houses were covered with shades either from the neighboring buildings and/or an upper story (47) and the rest (5) with trees. About 40% of the houses faced South, and 25% of the houses were oriented eastward. This is a common house building concept regarding house orientation in Pakistan because of the dominating sun path and radiations during peak sun hours in summer. Owners of the rest of the houses, facing North and West, had no options to choose the orientation of their houses because of limited available space.

### 2.3. Temperature Measurements

The outdoor and indoor measurements of temperatures were needed to calculate the *discomfort index* for various experimental setups. The data collection setup for this study was specially designed to avoid data variability, underestimations, and dimension error [29]. For indoor dry-bulb temperature (*Td*) measurements, data loggers (HOBO^®^ Temp Data Logger UX100-001, HOBO^®^ Temp/RH 2.5% Data Logger (UX100-011) were installed in the participating household to record and download data with a special shuttle (HOBO^®^ U-Shuttle, U-DT-x) or with a laptop/tablet. All the loggers were validated in a climate chamber against a licensed calibrated data logger. The accuracy of temperature sensors was ±0.21 °C to stand out from 10 to 90% for humidity. Data loggers were installed 1.5 m high on walls having no windows and/or minimal vents. The loggers were also kept away from heat, cold, moist, and dry sources, including the kitchen, heater, lights, fan, and water cooler, as suggested by Nguyen et al. [30] and Smargiassi et al. [8]. The indoor data were collected from 1 April to 31 August 2016 (150 days), which are historically summer months in Pakistan.

The indoor wet-bulb temperature (*Tw*) was measured using Kestrel 4600 Heat Stress Meter (Extreme Meters C/O Weather Republic, LLC. 3947 W Lincoln Hwy Unit 304 Downingtown, PA, USA) during the fortnight visits of each of the 52 houses, made for downloading the indoor *Td* data from HOBO^®^ data loggers, during five months of the study period. The Kestrel device measures *Tw* using an externally mounted, hermetically sealed thermistor sensor. The device was calibrated according to the manufacturer’s instructions, and data were recorded after the meteorologic values stabilized in about 6–7 min [31].

For outside weather data, an automated weather station (Wireless Vantage Pro2 Plus, Davis Instruments, Hayward, CA, USA) that included radiation sensors and a daytime fan-aspirated radiation shield for the temperature and humidity devices was installed ad hoc within the old campus of the Government College University, Faisalabad for real-time outdoor weather information of areas mimicking city center’s paved roads and surrounding buildings. Late delivering, installation of the weather station, and downloading issues with correct time-series data, delayed the outdoor data collection by 17 days. Therefore, the outdoor data became available from 17 April to 31 August 2016 (134 days).

### 2.4. Data Analysis

#### 2.4.1. Discomfort Index

Discomfort levels experienced by people were evaluated using the *discomfort index* formula introduced by Sohar et al. [32] and used by Ren et al. [33]. Sohar’s method compares the cumulative *discomfort index* (CDI), and cumulative effective temperature (CET) obtained from meteorological data [33]. This method is based on the sums CDI and CET values for each hour of the day and night, and modification of the Thom [34] method given below.
*Discomfort Index* = 0.4 (*Td* + *Tw*) + 15(1)

The formula of *discomfort index* designed by Thom [34] is for *Td* and *Tw* measured in °F. According to this formula, the *Td* and *Tw* are “equally important in respect to man’s feeling of discomfort in heat” therefore, *Td* and *Tw* measured in °C can represent *discomfort index* calculated as:*Discomfort Index* = (0.5 × *Td* + 0.5 × *Tw*)(2)

The formula of Equation (2) used in this study is based on the observations of Thom [34] that the changes in the sum of *Td* and *Tw* follow a similar pattern as changes in the effective temperature under the same conditions. The advantage of using Equation (2) is its advantage over Equation (1) that the former does not necessitate the measuring of the wind velocity, which is necessary for the estimation of the effective temperature [32].

#### 2.4.2. Time-Series Data

The time-series outdoor *Td* and *Tw* data were available from the weather station installed ad hoc for this study. The time-series indoor *Td* data were available from HOBO^®^ data loggers installed in each household. However, the time-series indoor *Tw* data were not available as it was only measured fortnightly instead of daily *Tw* separately for days and nights as mentioned in Section 2.3. Therefore, for time-series indoor *Tw*, two assumptions were made. First, to use representative *Tw* calculated from the regression model of the Engineering ToolBox [35] drawn as Figure 4a. Second, to use representative averaged indoor *Tw* calculated from outdoor *Tw-Td* relationship (Figure 4b). These assumptions were based on the literature reporting the use of representative data and assumptions adopted by Oreszczyn et al. [36], who evaluated mold (mould) and winter indoor relative humidity in low-income households in England and used the representative values of vapor pressure access from outdoor temperature. Similarly, Künzel [37] calculated a representative indoor relative humidity from moisture load using a developed relationship between the two variables with the cautionary statement that the estimated values correspond only to mean conditions and do not show the possible ranges. Once determining that since the tendency of the linear relation between moisture load and outdoor temperature of one location was obvious, they used the regression equation of the individual values to produce a summary of analog regression lines of all investigated homes. Representative values and assumptions have been commonly used in the literature of indoor/outdoor temperature and relative humidity/water vapor [38,39,40,41].

The validity of these assumptions was confirmed from the *discomfort index* values calculated using the measured indoor *Td*, *Tw* data and compared with the *discomfort index* values from indoor *Td*, indoor *Tw* calculated from the two regression models of Figure 4 (Table 2). The average *discomfort index* calculated from the measured indoor *Td*, *Tw* data deviated 1.44 °C from its mean in comparison with 1.63 and 2.24 °C from the respective means Figure 4a,b models. Based on a smaller deviation from the mean and the published nature, the Figure 4a relationship was used to calculate the missing values of indoor *Tw* from the respective indoor *Td*.

The risk of heat stress is experienced as mild or no discomfort for *discomfort index* below 24 °C, a moderate level of discomfort is experienced for *discomfort index* in the range of 24–28 °C, and *discomfort index* above 28 °C is considered as the severe level of discomfort [32].

### 2.5. The Interviews and Survey Questionnaires

Besides data collection from data loggers installed in houses of the five low-income communities, this study involved collecting information from the participating communities through interviews and survey questionnaires. All the 292 people of the 52 households were interviewed. The age group and gender of the interviewee are given in Table 3. Since over 90% of the participants were not literate, the survey questionnaires were filled in by the researchers from replies of the interviewees to various questions. The survey questions were about the sufferings of the people during summer days and the mitigation strategies adopted to combat heat-related health complaints. Children who were unable to express their feelings were represented by their mothers or elder siblings. The heat mitigation strategies were meant for adaptation to the changes specific to their sufferings during summer 2016.

## 3. Results

### 3.1. Difference between Indoor and Outdoor Temperatures

The indoor night-time *discomfort indices* were higher during May and June than during the rest of the months of the study period (Figure 5a). There were similar trends in the variations of outdoor *discomfort indices* (Figure 5b). The indoor *discomfort index* line remained farther below its respective *Td* line because of the big difference between the *Td* and *Tw* throughout the summer period (Figure 5a). Contrary to this, the outdoor *discomfort index* line remained closer to its respective *Td* line because of the smaller difference between the *Td* and *Tw* throughout the summer period as compared with the indoor lines (Figure 5b). A comparison of indoor and outdoor diurnal *discomfort indices* showed that the two indices lines overlapped each other during April and May, but the indoor *discomfort index* line remained below the outdoor *discomfort index* line for the rest of the summer months (Figure 5c). The increases in *discomfort indices* during April, May, and June were partly because of the historically prevailing continued drier conditions in Faisalabad during these months and the months before this period (Figure 2) than during July 2016, which received a record 178 mm of cumulative rain (Figure 5d) that might have resulted in lowering of *discomfort indices* during July and August 2016 both indoor and outdoor.

The effects of dry and wet conditions were substantial on indoor *Td*, *Tw*, and the resultant *discomfort indices* than on outdoor *discomfort indices* (Figure 6). The effects were greater on day and night indoor temperatures than on day and night outdoor temperatures (Figure 6b,d, respectively), as shown by greater offsetting of *Td* by *Tw* (Figure 5a,c, respectively). The effect of July rainfall was obvious on the squeezed distance between outdoor day and night *Td*, *Tw*, and *discomfort indices* (Figure 6b,d, respectively). Averaged over the five summer months of the study period, the daytime outdoor values of *discomfort indices* remained in the range of severe level of discomfort (>28 °C), while the night-time indoor and outdoor, and daytime indoor values of *discomfort indices* were in the top-end of mild discomfort levels (24–28 °C).

### 3.2. Daily and Hourly Calculated Discomfort Indices

Since May data reflected the highest *discomfort index*, further analysis of this dataset was conducted to evaluate variations in discomfort levels experienced by people during this month. Analysis of the daily variations revealed that the first three days of this month (1–3 May 2016) and the last week (21–28 May 2016) had the maximum *discomfort index* regardless of the design concepts of buildings (Figure 7). The data analysis further showed that direct exposure to sunlight caused a higher *discomfort index* than the houses having shades of trees or from the neighboring buildings (Figure 7a). There was a substantial difference between the *discomfort index* values of the shaded and non-shaded houses during the middle of the month (10–20 May 2016). The rest of the high *discomfort index* days did not have such a substantial difference. However, the difference between the *discomfort index* values of the ventilated and non-ventilated rooms was observed throughout May 2016, while the former type of rooms had a lower *discomfort index* than the *discomfort index* of the latter (Figure 7b).

The discomfort level experienced by people of the study houses touched its peak on 23 May 2016. Analysis of the diurnal variations calculated for 23 May 2016, reflected that the hours from 13:00 to 16:00 on this day had the highest *discomfort index,* and the morning hours of 07:00 to 09:00 and midnight durations of 21:00 to 00:00 had the lowest and medium *discomfort index* values, respectively, for all types of house building design concepts (Figure 8).

Almost similar effects of the house building design concepts were observed when hourly collected data were analyzed for the effect of house exposure to sunlight (Figure 8a), and house ventilation (Figure 8b). The houses exposed to sunlight and rooms without vent had higher values of *discomfort index* than the *discomfort index* of rooms of their opposite respective design concept.

### 3.3. Mitigation Strategies

Analysis of the survey questionnaire data revealed that the orientation of the house to the angle of the sun was used by the people to set their living habits and rules. For example, to ensure a living comfort, the people, especially females, preferred to live inside a house during different timings of the day. In addition to taking typical custom heat mitigation measures, including taking regular showers, drinking plenty of water, spending summer nights outside on lawns or the rooftops, and remaining under the tree shades outside the homes during peak hot hours. Under the cultural conditions, males of the community had the advantage to spend the peak hot hours outside their home under the shades of available trees. This benefit was not available to the female community.

With outdoor angles for sunlight, different parts of the house or a particular room had advantages and/or disadvantages for the people. Based on the sunlight angle, certain vents, windows, and doors remained shut or open by people to combat heat at varying hours of the day. When high temperature followed a daily peak-hour power cut, health issues arose, including diarrhea, dehydration, headache, and/or fever for both genders (Table 4).

Some popular heat mitigation strategies adopted by the five participating low-income communities during various heat-related health complaints were extracted from the analysis of the survey interviews and questionnaires. The strategies were gender-biased and had medical, cultural/customary backgrounds. For example, about 52% of the males and 28% of the females drank more water during dehydration, diarrhea, and eye infection. Over 11% and 19% of the males and females, respectively, moved to cooler places during fever. About 43% of the males and 51% of the females took water showers and rested to combat flu (runny nose), headache, and nosebleed. The people did not know how to cure muscular fatigue, skin allergy (from a type of Milia), and mild temperature.

## 4. Discussion

This study used meteorological, socio-economic, and health-related data collected from five low-income communities in Faisalabad’s neighborhood. Analysis of the meteorological data revealed that summer 2016 was harsh during the first two months when the indoor discomfort level was higher than the outdoor discomfort level dictating the neighborhood of the low-income families in Faisalabad spend their nights in cooler places outside their rooms (either in lawns or on the rooftops) to look for a little comfort against heat.

Rooms under the shades of trees or in the shades of neighboring buildings that did not have direct exposure to the sunlight had a lower discomfort level than the house without shades. Several environmental variables influence environmental conditions, including ambient air temperature, relative humidity, sunlight, wind direction, as well as speed, and solar radiation. These environmental conditions dictate the thermal comfort of an area. Additionally, the neighborhood of a house provides thermal comfort to the people of the house. Shades from the neighboring buildings and the trees improve thermal comfort, especially during summer [42,43]. Literature suggests that indoor thermal comfort can be achieved without using a mechanically or electrically build cooling system if the indoor temperature remains within the range of mild to moderate discomfort levels [44]. One of the discussed options is the use of specific types and layers of building walls that can play a significate role in regulating indoor air temperature [45,46]. Therefore, the application of appropriate materials for efficient building envelope design would be an effective alternative to maintain mild to moderate discomfort levels for low-income communities of cities like Faisalabad.

The study results are in concurrence with the findings of Astrom [47], who also reported sensitivity of indoor temperature to outdoor temperature primarily based on house building design concepts, outdoor conditions of a locality, and the outdoor temperature itself. However, house design concepts were related to old wisdom-based construction protocols; for example, well-ventilated high-ceiling, and shaded rooms either with trees, the neighboring buildings, or by an upper story construction were able to mitigate outdoor temperature in this study.

The amount of heat transferred through the walls and rooftops of the building made from various materials depends on the conduction, convection, and radiation capacity of the building materials [48]. During the daytime, the solar radiation hits the surfaces of external walls and rooftops. A part of the is released to the outdoor environment, and the other part is absorbed and conducted across the material. Theoretically, the inner surface of the walls or the rooftop, exposed to the outdoor temperature, exchange heats with the room air and other objects inside the room through convection and radiation. These two heat transfer methods regulate the indoor air temperature and consequently influence the level of discomfort inside the room. The heat exchange rate depends on the solar radiation, indoor temperature, outdoor temperature, material thermophysical properties, and exposed surface area.

Information collected from the survey questionnaire showed that people of the low-income areas of Faisalabad followed various mitigation strategies to combat heat discomforts depending upon the gender orientation of the people who face higher heat indoor during the first few months of summer. Large variation exists in indoor thermal comfort according to different climates, times of the year, and cultures [23]. While during night-time, the night indoor temperature remained higher than the outdoor usual temperature emphasizing the need to adapt to the challenging conditions of spending nights outside their rooms.

In addition to taking typical custom heat mitigation measures, including taking regular showers, drinking plenty of water, spending summer nights outside on lawns or the rooftops, and remaining under the tree shades outside the homes during peak hot hours, planting trees in an area, and developing open parks with greenery and thick canopy trees can be beneficial for a community. These responses and the actions and the lifestyles and beliefs ensure that people could survive in almost all the wide variety of conditions found across the planet [23].

## 5. Conclusions

Heat warnings are typically based on outdoor conditions, where the people usually consume little time, especially during the hottest periods or the periods when the warnings are issued. Our conclusions on seasonal heat stress levels in low-income homes show a true picture of actual conditions in the urban residential environments of most of the South Asian cities. Poor communities are most affected because they live in crowded, built-up urban neighborhoods, which are hot and humid, and not equipped with air conditioners. Discomfort levels in low-income houses can be further lessened with some economic and efficient management practices, including using natural ventilation, roof and wall insulation, and shading houses with trees. The use of evaporative coolers, smart and/or high roof and wall with cross-ventilation options can dramatically reduce indoor temperatures and sufferings of the people. Caution may, therefore, be practiced when using passive cooling techniques. Since a substantial fraction of homes exceeds heat thresholds during the extreme events in Faisalabad, there is an immediate need for greater understanding and monitoring of urban heat in cities at risk of heat stress. The findings of this study are also applicable to the people of low-income areas of other parts of the country, such as Lahore and Rawalpindi, that resemble the conditions of Faisalabad. These people may face severe discomfort, due to arid to semiarid climatic conditions where temperature plays a major role in heat stress thresholds. In addition to taking typical custom measures, including taking showers, spending summer nights outside in lawns or on the rooftops, and remaining under the shades during summer days, planting trees in an area can be beneficial for the community in addition to building open community parks with an abundance of greenery and occupancy of thick-canopy trees.

## Figures and Tables

**Figure 1 ijerph-18-02535-f001:**
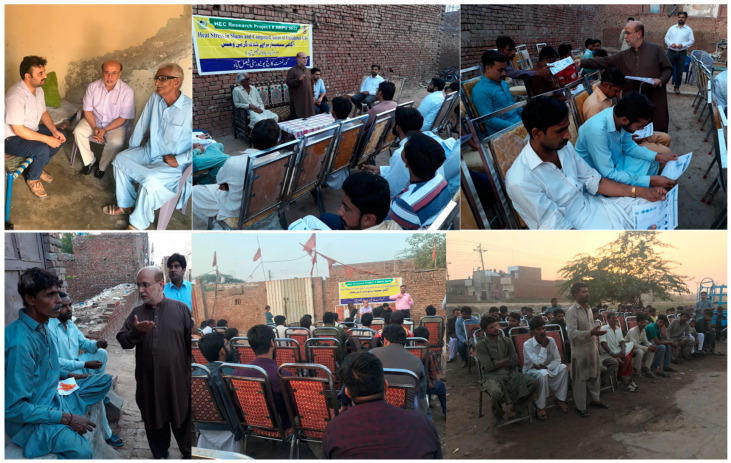
Meeting of some members of study team with community heads to convince them for attending community awareness seminars in Babuwala and Muradabad sites of this study about the importance of the talk and their participation in the study activities.

**Figure 2 ijerph-18-02535-f002:**
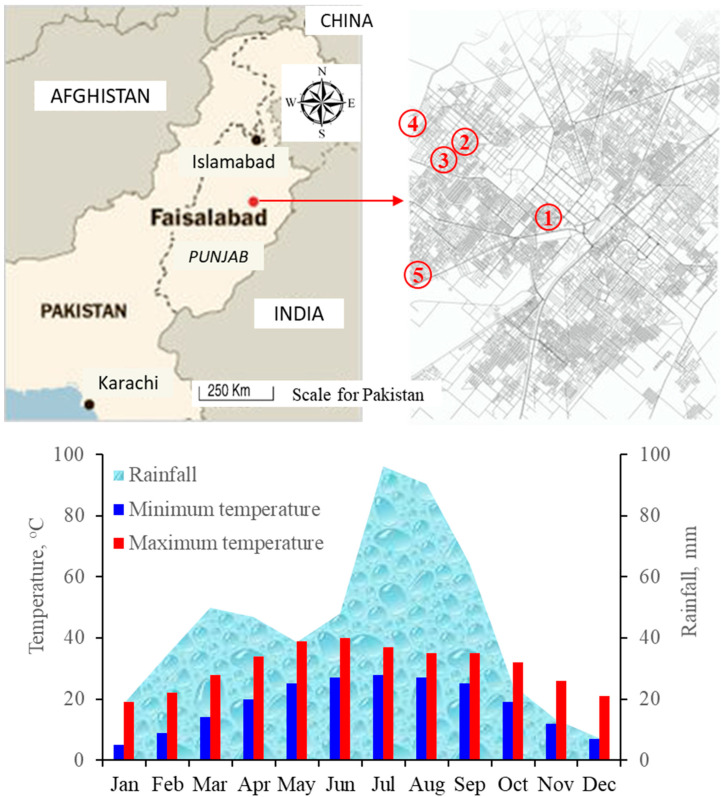
The geographic location of Faisalabad city and the five low-income communities (top) the past 50-year record of monthly means for rainfall, minimum temperature, and maximum temperature for the Faisalabad region.

**Figure 3 ijerph-18-02535-f003:**
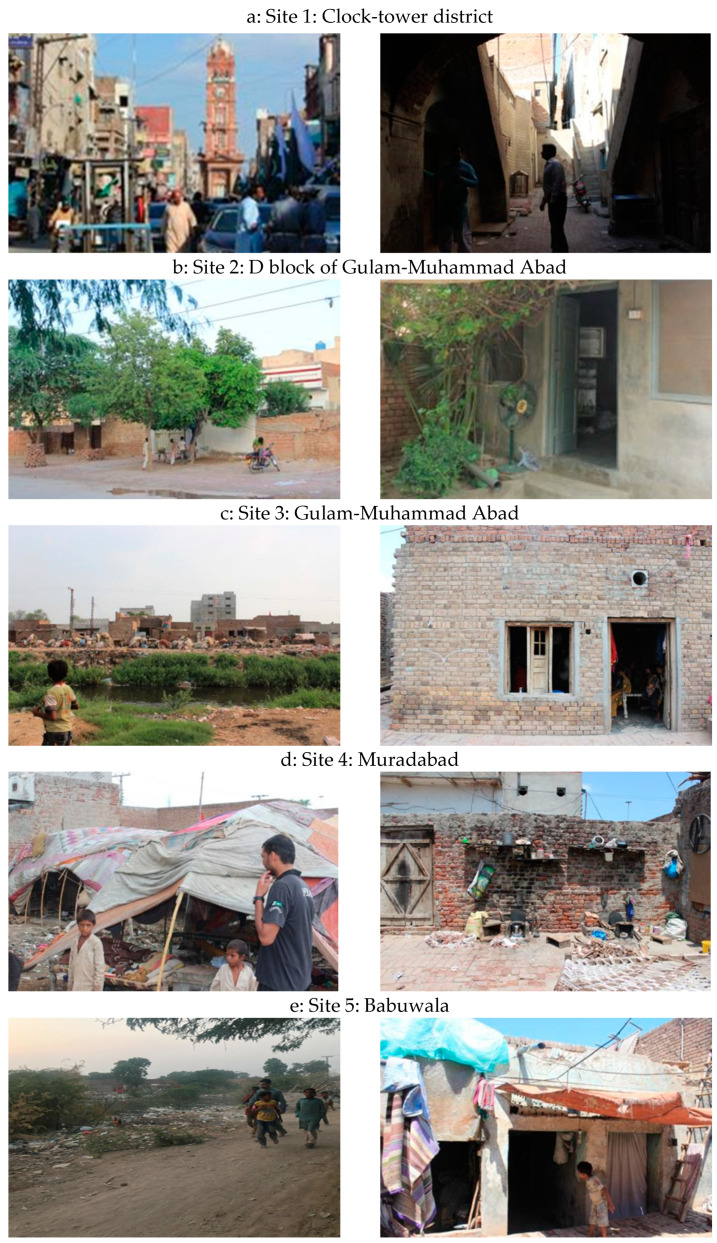
Snapshots of the neighborhood (**left**) and indoor conditions (**right**) of (**a**) Site 1: The clock tower district; (**b**) Site 2: D block of Gulam-Muhammad Abad; (**c**) Site 3: Gulam-Muhammad Abad; (**d**) Site 4: Muradabad; and (**e**) Site 5: Babuwala.

**Figure 4 ijerph-18-02535-f004:**
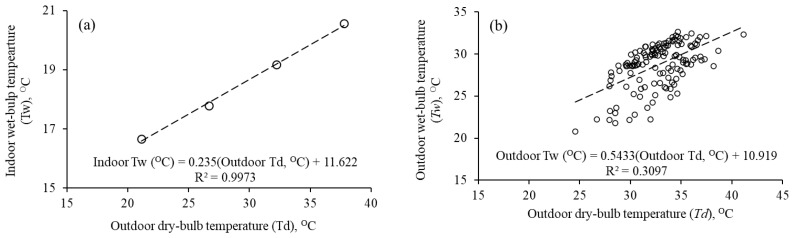
Relationships between (**a**) indoor wet-bulb and outdoor dry-bulb temperatures drawn from the data of the Engineering ToolBox [35], and (**b**) between the observed outdoor wet-bulb and outdoor dry-bulb temperatures.

**Figure 5 ijerph-18-02535-f005:**
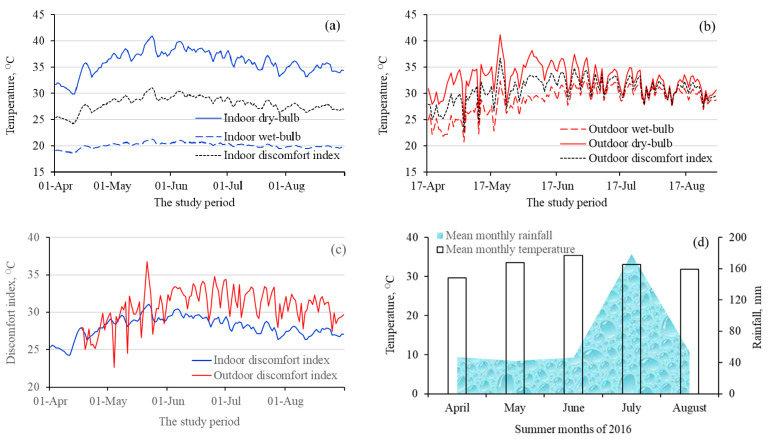
Diurnal wet-bulb temperature, dry-bulb temperature, and *discomfort indices* for (**a**) indoor (**b**) outdoor, and (**c**) comparison of indoor and outdoor diurnal *discomfort indices*, and (**d**) monthly averages of daily mean temperatures (plotted on the primary *y*-axis) and cumulative monthly rainfall (plotted on the secondary *y*-axis) for the study period.

**Figure 6 ijerph-18-02535-f006:**
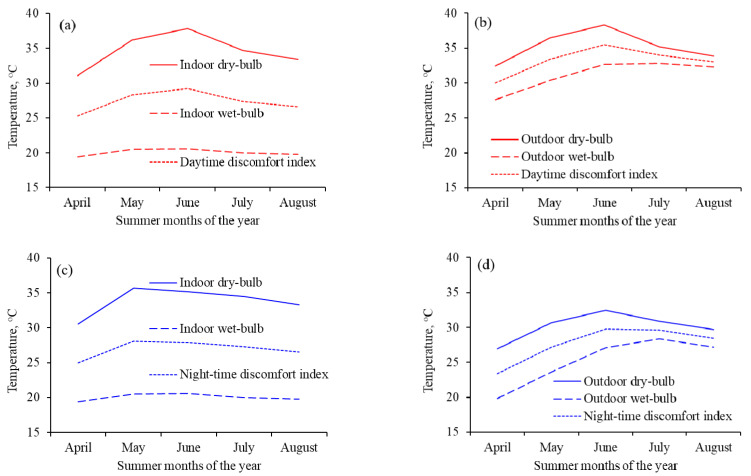
Variations in the monthly averaged *discomfort indices* caused by increasing and/or decreasing dry- and wet-bulb temperatures of the indoor day, night (**a**,**c**), and outdoor day, night (**b**,**d**) conditions during the five months of summer 2016.

**Figure 7 ijerph-18-02535-f007:**
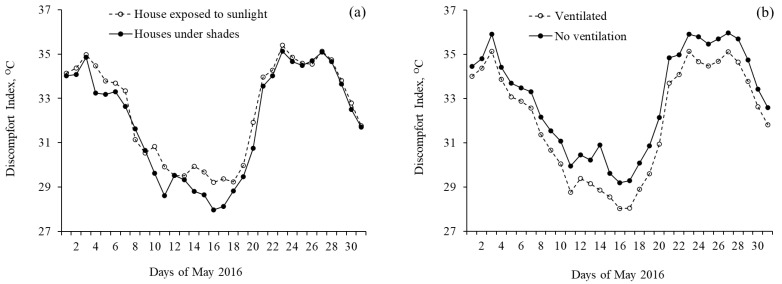
Discomfort indices for the days of the highest *discomfort index* month (May 2016) of the study period based on the effects of house building design concepts, including (**a**) houses exposed to sunlight versus shaded houses, and (**b**) houses with and without proper ventilation.

**Figure 8 ijerph-18-02535-f008:**
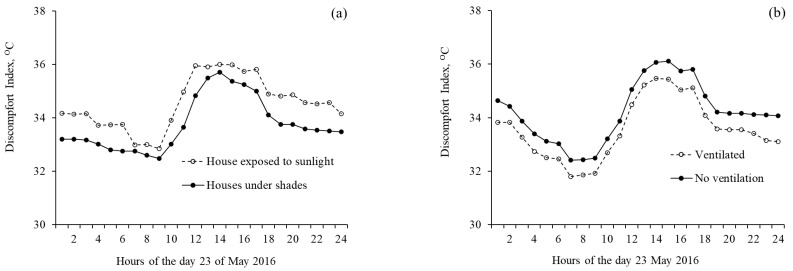
Diurnal effect of the house building design concepts in terms of discomfort indices for 23 May 2016, when people faced the highest discomfort level during summer 2016 under the scenarios of (**a**) houses exposed to sunlight versus shaded houses, and (**b**) houses with and without proper ventilation.

**Table 1 ijerph-18-02535-t001:** Description of variables and summary statistics of socio-economic conditions and building design concepts of houses (*n* = 52) considered in this study.

Category	Households and Building Design	Statistics
**Household ratios**		
Family size	Number of dwellers	6.6/house
Gender proportion	Female:Male	~1.0 (148:144)
Crowding per household	No rooms/No of occupants	0.4 (114/292)
**Building design concepts**		
Rooms with windows	With	61%
	Without	38%
	One window	57%
	Two windows	4%
Rooms with vents	With	50%
	Without	50%
	One vent	32%
	Two or more vents	18%
Rooms with shades	With	89%
	Without	11%
	Fully shaded	59%
	Partly shaded	30%
**Necessities**		
Home appliances	Evaporative water cooler	9.6%
	Fan	100%
	Fridge	55%
	UPS/Generator	10%
	Own toilet	87%
	Separate indoor kitchen	66%
	Washing machine	53%
Experience power outage		100%

**Table 2 ijerph-18-02535-t002:** Values of *discomfort index* calculated from measured indoor dry-bulb temperature (*Td*) and (**a**) measured indoor wet-bulb temperature (*Tw*), (**b**) indoor *Tw* calculated with Engineering ToolBox [35] equation (Figure 4a), and (**c**) indoor *Tw* calculated with measured outdoor *Td*-*Tw* relationship (Figure 4b).

Date dd/mm	a. Discomfort Index from Measured *Td* and *Tw*, °C	b. Discomfort Index from Engineering TooBox Equation [35], °C	c. Discomfort Index from Measured Outdoor *Td*-*Tw* Relation, °C
01/04	28.2	25.4	29.9
16/04	28.2	25.2	29.7
01/05	31.5	28.8	34.2
16/05	29.2	28.8	29.7
01/06	31.7	29.4	34.9
16/06	32.1	29.6	35.2
01/07	31.6	29.4	34.9
16/07	30.3	27.6	32.6
01/08	30.9	27.5	32.5
15/08	30.2	27.0	32.0
Minimum, °C	28.2	25.2	29.7
Maximum, °C	32.1	29.6	35.2
Mean, °C	30.4	27.9	32.6
SD, °C	1.44	1.63	2.24

SD, standard deviation.

**Table 3 ijerph-18-02535-t003:** Age group and gender of the interviewees who were interviewed for their experience about heat-related complaints and mitigation strategies.

Gender	Adult: ≥18 Years Old (% of Total Participants)	Non-Adult: <18 Years Old (% of Total Participants)	Total Participants (%)
Female	85 (30.1%)	63 (21.6%)	148 (50.7%)
Male	76 (26.0%)	68 (23.3%)	144 (49.3%)
Total	161 (55.1%)	131 (44.9%)	292 (100%)

**Table 4 ijerph-18-02535-t004:** Gender-wise (*n* = 104 @ 2 samples per household) data for heat-related health complaints and heat mitigation strategies adopted by the interviewees.

Health Complaints	Percent			Heat Mitigation Strategies	Percent		
	Male	Female	All *		Male	Female	Total
Dehydration	16.98	14.89	15.9	Drank more water	21.28	13.21	17.25
Diarrhea	26.42	17.02	21.7	Drank water and took a shower	14.89	5.66	10.28
Eye infection	—	4.26	2.13	Drank water and rested	14.89	9.43	12.16
Fever	24.53	23.4	24.0	Moved to cooler places	10.64	18.87	14.8
Flu (runny nose)	1.89	6.38	4.14	Took more than usual rest	10.64	7.55	9.10
Headache	16.98	27.66	22.3	Took multiple showers a day	21.28	41.51	31.4
Nosebleed	1.89	6.38	4.14	Took shower and stayed home	6.38	3.77	5.08
Muscular fatigue	1.89	—	0.95	Did not know what to do	—	—	—
Skin allergy	5.66	—	2.83	Did not know what to do	—	—	—
Mild temperature	2.77	—	1.39	Did not know what to do	—	—	—

* percentage calculated based on all genders’ data.

## Data Availability

This study did not report any data.

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
