# Peer review of "Thermal Discomfort Levels, Building Design Concepts, and Some Heat Mitigation Strategies in Low-Income Communities of a South Asian City"

_ijerph, 2021, doi:10.3390/ijerph18052535_

Round 1
Reviewer 1 Report
The article " Indoor heat discomfort effects of building attributes in low-income neighborhoods of a South Asian city and the heat mitigating strategies" evaluates the effects of house attributes on building up of discomfort level in a city area of Pakistan, which hosts many low-income communities. It demonstrates the study of outdoor and indoor weather data collected from April to August 2016 in urban settings and the 52 houses. The discomfort index values related to the building attributes, including house exposure to sunlight and house ventilation, were calculated from wet-bulb and dry-bulb temperature.
Generally, the research method presented in the paper seems to be not justified enough. There is no clear why the Authors have chosen the experimental results for the analysis. Also, there is no information on how exactly the received oversized discomfort index could be proved. Even though the study identified some popular heat mitigation strategies commonly adopted by low-income areas, the proposals were not tested nor confirmed as effective. Finally, the conclusions are very general, not confirmed by the experiments, and don't give practical solutions to eliminating the problem of overheated houses in Pakistan, where the problem with low incomes has to be taken into account.
To summarising, the paper doesn't present a level required for the IJERPH. It could be improved by presenting additional research on how several heat mitigation strategies decrease the received discomfort index and how high costs would be expected for their implementation.
Some detailed comments and remarks are listed below:
The "semi-arid" word is written as "semi-arid" and as "semiarid" – it should be consistent;
Page 3 - The description of the 5 neighborhoods taken for the experiments could be completed with some photos;
Figure 1 – it should be "INDIA" not "INDA"
Figure 2 – the data is from 2016 only?
Page 6 – the discomfort index should be described more clearly, and it should be justified why the equation is exactly like it is
Author Response
Please find the attached file for authors' response to your comments.

Reviewer 2 Report
The paper fits the scope of the journal.
The introduction describes the background and the research question is clear, presenting this work as a case study to evaluate the effects of the selective house attributes on building up of discomfort level conditions in low-income areas of Faisala-bad, Pakistan.
The case study is well-described in section 2. The case study was carried out through a mix of qualitative (interviews) and quantitative methods (Statistical and measuraments).
By doing so, the risk of heat stress was evaluated by measuring the discomfort index, whose definition was provided.
Results are focused on three aspects: Difference between indoor and outdoor temperatures; Daily and hourly calculated discomfort indices; Mitigation strategies. The image are readable.
To sum up, I think that this work is interesting and well organised. However, I do not think that this paper proposes innovative methods. By contrast, I considers this geographical application relevant in order to support the dissemination of appropriate technologies and practices.
So, well done.
Author Response

(The authors gave the same response as above.)

Reviewer 3 Report
Good work.
It could be important to obtain more apparatus in order to apply your methodology to your own data.
Possible extent to other parts of the World with similar climate (Koppen –Geiger classification).
I am not clear (page 6) if "mold" should be "mild"
Author Response

(The authors gave the same response as above.)

Reviewer 4 Report
Please, see file attached.

Author Response

(The authors gave the same response as above.)

Round 2
Reviewer 1 Report
I confirm that the paper has been improved by presenting additional research on how several heat mitigation strategies decrease the received discomfort index and how high costs would be expected for their implementation what was suggested in my original review. Also, my detailed comments are included in the new version. To summarising, the paper after the corrections could be published in IJERPH.
Author Response
Please see the attached response file.
Thanks,

Reviewer 4 Report
Please, see attached file.

Author Response

(The authors gave the same response as above.)
